# Nutritional Adequacy and Patient Perceptions of the Hospital Inpatient Haemodialysis Menu: A Mixed Methods Case Series

**Kate Neal** [1,2,*], **Fatima Al Nakeeb** [3] **and Kelly Lambert** [3]

1 Nutrition and Dietetics Department, Liverpool Hospital, Sydney, NSW 2170, Australia
2 Renal Department, Liverpool Hospital, Sydney, NSW 2170, Australia
3 School of Medical, Indigenous and Health Sciences, University of Wollongong, Wollongong, NSW 2500, Australia; fatima.alnakeeb@health.nsw.gov.au (F.A.N.); klambert@uow.edu.au (K.L.)
* Correspondence: kathleen.neal@health.nsw.gov.au; Tel.: +61-0472801187

**Abstract:** Aim: To evaluate the nutritional adequacy of the hospital haemodialysis menu, quantify the dietary intake of hospitalised haemodialysis patients and explore patient perceptions of the menu. Methods: The menu analysis compared the default menu to reference standards using a one sample *t*-test via SPSS. Eight hospitalised haemodialysis patients were purposively interviewed using semi-structured interviews. Thematic analysis was used to identify the dominant themes. The participant's actual dietary intake was calculated and compared to individual nutrients using evidence-based guidelines. Results: Compared to the reference standards, the default inpatient haemodialysis menu did not provide adequate energy ($p < 0.001$, mean = 8767 kJ/day $\pm$ 362), sodium ($p < 0.001$, mean = 72 mmol/day $\pm$ 9), potassium ($p < 0.001$, mean = 64 mmol/day $\pm$ 4), vitamin C ($p \leq 0.001$, mean = 33 mg/day $\pm$ 10) and fibre ($p < 0.001$, mean = 26 g/day $\pm$ 3). Inadequate intake of energy and protein occurred in half of the participants. Passive acceptance of the menu, environmental and cultural considerations contributed to missed food opportunities impacting the patient experience and limited intake. Conclusions: The profile of the current default inpatient haemodialysis menu impacts the dietary intake and the experience of haemodialysis inpatients. It is recommended that the default menu is optimised in line with evidence-based guidelines for inpatients.

**Keywords:** haemodialysis; dietary intake; inpatient; qualitative research





## 1. Introduction

The management of chronic kidney disease (CKD) is hindered by the complexity of the nutrition prescription [1]. Patients undergoing haemodialysis are confronted with a complex and contradictory range of dietary restrictions in coexistence with heightened nutritional needs [2]. Haemodialysis treatment requires an individual to modify their diet to meet their additional needs for energy, protein, iron, zinc, folate and vitamin C, whilst decreasing potassium, phosphorus, sodium and fluid [3–6]. Deviating from these recommendations has important clinical implications including malnutrition, fluid overload, hyperkalaemia and cardiovascular disease [1]. Protein–energy malnutrition in CKD is estimated to affect 46% of haemodialysis patients whilst increasing the risk of morbidity and early mortality [7–9]. An individual's nutritional status may be further compromised by symptoms associated with CKD and haemodialysis treatment including poor appetite, nausea and fatigue [3,10].

The Kidney Disease Outcomes Quality Initiative (KDOQI) guidelines outline the most recent evidence-based nutritional recommendations for macro and micronutrients for patients at all stages of CKD [11]. These guidelines were updated and released in 2020 using evidence available up to April 2017. Despite patients and caregivers understanding the importance of adequate nutritional intake in the context of CKD, compliance to dietary recommendations remains an issue in outpatient and inpatient settings [12,13]. A study

conducted in Singapore completed semi-structured interviews to understand the perceptions of 14 inpatients regarding their adherence to the recommended dietary and fluid restrictions for haemodialysis [12]. They found that participants viewed these restrictions as a burden and therefore would experiment with these restrictions until adverse effects were evident. This study did not assess the nutritional adequacy of participant's intake or the menu provided.

Nutritional intake in the inpatient setting is known to be suboptimal due to factors such as disease burden, access to food and missed meals [14]. It has been identified that 85% of inpatients receiving therapeutic diets are unable to meet their nutritional requirements from the inpatient menu provided [15]. However, this study excluded haemodialysis patients and there remains a knowledge gap regarding whether the haemodialysis menu is meeting the nutritional needs of inpatients and the potential barriers to oral intake in the inpatient setting. It is likely that oral intake is further compromised within the population of haemodialysis inpatients due to the significant dietary restrictions and limited food options associated with the inpatient haemodialysis menu [2]. In our setting, the inpatient haemodialysis menu is highly restrictive and designed to limit the intake of potassium, phosphate, salt and fluid at each meal and mid-meal. This prevents patients from ordering above the prespecified nutrient limits. The haemodialysis inpatient menu is derived from the full diet and any non-compliant options are removed meaning that the haemodialysis menu has fewer options to select from than the full diet. The current inpatient haemodialysis menu has not undergone any recent analysis against the revised nutrition guidelines released in 2020 by KDOQI [11].

It is evident from the literature that oral intake among inpatients is often suboptimal and that those receiving haemodialysis are at an even higher nutritional risk due to the complexity of the renal diet prescription—especially with the limitations of the inpatient menu system, increased catabolic reactions and potential symptom burden associated with haemodialysis [2,11]. However, few studies have examined the oral intake of inpatients receiving the haemodialysis menu and haemodialysis treatment, or the perceptions of these patients about the restrictive nature of the menu. Understanding the patients' perspectives will help to inform changes to improve the quality of their care [16]. Therefore, this mixed method case series aimed to evaluate the nutritional adequacy of the inpatient haemodialysis menu, quantify the nutritional intake of haemodialysis inpatients and explore patient perceptions of the haemodialysis menu.

## 2. Materials and Methods

A mixed methods approach was adopted due to its ability to explore complex relationships between quantitative and qualitative data [17]. It comprised three phases: (i) a menu analysis; (ii) quantification of inpatient intake and (iii) qualitative analysis of patient perceptions about the haemodialysis menu.

This project was approved by the Research Ethics and Governance Information System at [removed for peer review purposes] Hospital (Approval number 2021/ETH00125). This hospital is a large (855 bed) metropolitan teaching hospital in [removed for peer review purposes], which hosts a large culturally diverse population, and meal provision is arranged by private contractors to the public health system, HealthShare. HealthShare utilise an electronic menu ordering system called CBORD, with default menu options provided for those unable to order meals independently. The default haemodialysis menu is on a 14-day menu cycle.

The first phase of the study involved analysis of the default items provided on the haemodialysis menu. The therapeutic diet is intended to provide 9500 kJ, 90 g protein, <100 mmol sodium, <70 mmol potassium and <1200 mg (38 mmol) phosphate per day [18]. The default menu is designed to theoretically meet the needs of a 76 kg male [18]. The average daily amounts provided on each day of the 14-day haemodialysis menu cycle were sourced and entered into SPSS (version 25; IBM Corporation, Chicago, IL, USA) for energy, protein, fibre, sodium, potassium, phosphate, iron, zinc, folate and vitamin C. Calculations

are outlined in Table 1 and were based on the KDOQI guidelines [11] for energy (126 kJ/kg), protein (1.2 g/kg), saturated fat (<7% of energy intake), carbohydrate (50–60% of energy intake), sodium (<100 mmol/day) and potassium (1 mmol/kg); the Nutrient Reference Values for fibre (30 g/day), iron (8 mg/day), zinc (14 mg/day), folate (400 µg/day), and vitamin C (45 mg/day). A phosphate restriction of 32 mmol/day was applied [19]. A one sample $t$-test was conducted using SPSS (version 25; IBM Corporation, Chicago, IL, USA) to compare the nutrient profile of the average provision of the haemodialysis default menu to the reference values of a 76 kg male according to the updated KDOQI nutrient guidelines [11].

**Table 1.** Nutritional provision of the haemodialysis menu compared to requirements for a 76 kg reference individual.

| Nutrient | Reference Value | Requirement for 76 kg Reference Individual | Average Haemodialysis Diet Menu Provision | $p$-Value |
|---|---|---|---|---|
| Energy | 126 kJ/kg | 9576 kJ/day | 8767 kJ $\pm$ 362 | <0.001 |
| Protein | 1.2 g/kg | 91 g/day | 90 g $\pm$ 6 | 0.40 |
| Sodium | <100 mmol/day | <100 mmol/day | 72 mmol/day $\pm$ 9 | <0.001 |
| Potassium | 1 mmol/kg | 76 mmol/day | 64 mmol/day $\pm$ 4 | <0.001 |
| Phosphate | 32 mmol/day | 32 mmol/day | 46 mmol/day $\pm$ 2 | <0.001 |
| Iron | 8 mg/day | 8 mg/day | 14 mg/day $\pm$ 1 | <0.001 |
| Zinc | 14 mg/day | 14 mg/day | 14 mg/day $\pm$ 2 | 0.46 |
| Vitamin C | 45 mg/day | 45 mg/day | 33 mg/day $\pm$ 10 | <0.001 |
| Folate | 400 ug/day | 400 ug/day | 439 mg/day $\pm$ 27 | <0.001 |
| Fibre | 30 g/day | 30 g/day | 26 g/day $\pm$ 3 | <0.001 |

In phase 2, recruitment occurred over 5 weeks between April 2021 (7 April 2021) and June 2021 (2 June 2021). As a result of COVID-19 hospital ward lockdowns, the study recruitment period was reduced to a total of 3.5 weeks. A total of eight participants were recruited via purposive sampling. The renal ward list of the hospital was screened to identify eligible participants who had received the haemodialysis diet for three or more consecutive days. This timeframe was selected to ensure that participants had received the menu on more than one occasion in order to provide useful insights for the study. Other inclusion criteria included that patients were English speaking, over 18 years of age, receiving haemodialysis treatment and able to provide informed consent to participate in the study. Exclusion criteria for the study included: patients who had not received the haemodialysis diet, patients who could not converse in English, are under 18 years of age, pregnant, not receiving haemodialysis treatment, patients also receiving a texture modified or allergy diet and those unable to provide informed consent.

The nursing staff were consulted regarding the cognitive state of potential participants and their ability to provide informed consent. Following the identification of potential participants, individuals were approached by the principal investigator and invited to participate in the study, if they met full inclusion/exclusion criteria upon questioning, and were presented with an information sheet, alongside a consent form. Once consent was obtained, the participant was allocated a study identification code. A total of eight participants were recruited to the study.

Recruited participants underwent an assessment at their bedside. These assessments included the Patient-Generated Subjective Global Assessment (PG-SGA©) to assess nutritional status and determine the presence of malnutrition in each participant [20], and a 24 h recall to determine dietary intake. Inpatient meal provision in our hospital utilises an electronic menu system (CBORD). The CBORD menu selections for each individual were compared to the 24 h recall data for corroboration of participant reports. Dietary intake for each participant obtained from the 24 h recall was quantified using the ready reckoner inbuilt within CBORD and food items consumed by the participants sourced from outside of the hospital were calculated using FoodWorks (version 9; AusFoods 2017, Xyris Pty Ltd.,

Highgate Hill, QLD, Australia). This provided an estimation for the daily intake of energy, protein, carbohydrate, fat, fibre, potassium, phosphate, sodium, calcium, iron, zinc, folate and vitamin C intake for each participant. This estimation of intake was then compared to the individual's requirements for each nutrient that was calculated utilising the KDOQI guidelines (for energy, protein, fat, carbohydrate, potassium and sodium) [11], the 2006 evidence-based guidelines (for phosphate) [19] and the 2006 Nutrient Reference Values [21] (for fibre using adequate intake data, iron using recommended dietary intake (RDI) data, zinc using RDI data, folate using RDI data and vitamin C using RDI data).

Phase 3 consisted of semi-structured interviews that were conducted by a member of the research team [initials removed for peer review purposes] at the patient's bedside. Participants were aware of the goals of the study and that the researcher was a student completing their degree. Interviews were scheduled at times indicated by the participant's preference and availability and the average interview length was approximately 45 min. The interviewer, participant and, if desired by the participant, a carer were present at the interviews. The interviews included several open-ended questions about the participant's perceptions of the haemodialysis menu, food items offered, barriers to intake and suggestions for improvement. The interview style allowed for open discussion and for participants to talk at length. Participants were aware that they could end the interview at any time. The interview questions are shown in Table S1 in the Supplementary Materials. These questions were developed by two experienced renal dietitians based on previous experience with inpatient food service management. Interviews with participants were digitally recorded and transcribed verbatim using Otter io software (https://otter.ai/, accessed on 15 June 2023). To ensure accuracy of transcription, the transcripts were verified by comparing these with the audio recordings and corrected if required. Participants were not invited to provide feedback on the transcriptions. Field notes were also kept by the researcher.

A simple thematic analysis of interview data was conducted. In brief, this involved the four steps of (1) data immersion, (2) coding, (3) creating categories, and (4) theme identification [22]. Corrected transcripts were uploaded to Dedoose software (Version 9.0.17, web application for managing, analysing, and presenting qualitative and mixed method research data (2021). Los Angeles, CA: SocioCultural Research Consultants, LLC (URL: www.dedoose.com, accessed on 15 June 2023)) in order to undertake the thematic analysis. The research team ($n = 3$) inductively identified concepts by line-by-line coding of the transcripts. All team members met to discuss and refine the themes and identify conceptual links between themes. Details of the study design and analysis are reported according to the COREQ guidelines for the reporting of qualitative research [23].

### 3. Results

A total of ten participants were eligible to participate in the study, however one participant declined, and one interview was cancelled due to COVID-19 restrictions. Therefore, interviews were completed as outlined in Figure S1 of the Supplementary Material. The characteristics of the participants are shown in Table 2. The average age of participants was 71 years (range 60–78 years). Seven of the eight participants were male. Of the eight participants, three were from an Anglo-Celtic background, three were Southeast Asian, one participant was Middle Eastern and one participant was European.

**Table 2.** Demographics of study sample.

|  | Study ID 1 | Study ID 2 | Study ID 3 | Study ID 4 | Study ID 5 | Study ID 6 | Study ID 7 | Study ID 8 |
|---|---|---|---|---|---|---|---|---|
| Gender | M | F | M | M | M | M | M | M |
| Age | 78 | 66 | 76 | 64 | 71 | 60 | 76 | 78 |
| Weight (kg) | 65 | 39.4 | 58 | 66 | 96 | 65 | 100 | 71 |
| BMI (kg/m$^2$) | 22.8 | 17.5 | 24.1 | 27.5 | 33.2 | 25.4 | 33.3 | 24.6 |
| PG-SGA score | A | B | B | C | B | C | B | C |
| Food provided from Home | Yes | Ad hoc | Yes | Nil | Nil | Nil | Nil | Nil |

Legend: Patient-Generated Subjective Global Assessment (PGSGA) score—A: well-nourished; B: moderate–mild malnutrition; C: Severely malnourished.

The menu analysis found that the default haemodialysis menu provided significantly less energy (Table 1, $p < 0.001$, mean provision = 8767 kJ/day $\pm$ 362, reference = 9576 kJ), sodium ($p < 0.001$, mean provision = 72 mmol/day $\pm$ 9, reference = 91 g), potassium ($p < 0.001$, mean provision = 64 mmol/day $\pm$ 4, reference = 76 mmol), vitamin C ($p$ = <0.001, mean provision = 33 mg/day $\pm$ 10, reference = 45 mg) and fibre ($p < 0.001$, mean provision = 26 g/day $\pm$ 3, reference = 30 g/day) than required for the reference individual. The menu analysis found that the haemodialysis menu provided significantly greater levels of phosphate ($p < 0.001$, mean provision = 46 mmol/day $\pm$ 2, reference = 32 mmol/day), iron ($p < 0.001$, mean provision = 14 mg/day $\pm$ 1, reference = 8 mg/day) and folate ($p < 0.001$, mean provision = 439 ug/day $\pm$ 27, reference = 400 ug/day) than required for the reference individual. Protein ($p = 0.20$, mean provision = 90 g/day $\pm$ 6, reference = 91 g/day) and zinc ($p = 0.23$, mean provision = 14 mg/day $\pm$ 2, reference = 14 mg/day) were not found to be significantly different from the reference range values. A full summary of results can be found in Table 1.

Phase two identified that most of the participants (87%) were malnourished according to the PG-SGA. Table 3 shows analysis of individual food intake. The average energy intake was 3818 kJ $\pm$ 1410 kJ/day (range 1087–5488 kJ) and the average protein intake was 44 g/day $\pm$ 21 g/day (range 11–74.5 g/day). Of the eight participants: one exceeded their estimated energy and protein requirements (EER, EPR); three exceeded > 50% of their EER and EPR; and four participants consumed <50% of their EER and EPR. Recommendations for saturated fat intake were exceeded by most participants (n = 5, range = 31–160%). None of the participants met the fibre recommendations, with the average intake being 26 g/day $\pm$ 5 g/day (range 6–17 g/day). Regarding mineral intake, no participant met the estimated requirements for potassium (mean intake = 29 mmol/day $\pm$ 11 mmol/day, range = 7–33 mmol/day), phosphate (mean = 20 mmol/day $\pm$ 9 mmol/day, range = 17–43 mmol/day), sodium (mean = 38 mmol/day $\pm$ 15 mmol/day, range = 16–58 mmol/day), calcium (mean = 344 mg/day $\pm$ 157 mg/day, range = 194–577 mg/day), zinc (mean = 6 mg/day $\pm$ 3 mg/day, range = 2–12) or folate (mean = 228 ug/day $\pm$ 83 ug/day, range = 86–310 ug/day). Four participants met the nutrient reference value for iron, with the mean intake being 7 mg ($\pm$ 3 mg/day). One participant met the vitamin C requirement, with the mean intake being 20 mg/day (mean = 22 mg/day, $\pm$ 14 mg/day, range 1–45 mg/day).

**Table 3.** Case series of dietary intake amongst haemodialysis inpatients compared to individual nutritional requirements.

| | | Study ID 1 | Study ID 2 | Study ID 3 | Study ID 4 | Study ID 5 | Study ID 6 | Study ID 7 | Study ID 8 | Mean | SD |
|---|---|---|---|---|---|---|---|---|---|---|---|
| Energy (MJ) | | Energy: 105–146 kJ/kg [11] | | | | | | | | | |
| | EER | 6.9–9.7 | 4.1–5.8 | 6.1–8.5 | 7.2–10 | 10–14.1 | 6.8–9.5 | 10–13.9 | 6.8–9.5 | | |
| | Actual | 5.5 | 4.5 | 2.6 | 5.1 | 3.9 | 1.1 | 3.8 | 4.2 | 3.8 | 1.4 |
| | % | 80 | 100 | 43 | 71 | 39 | 16 | 38 | 62 | | |
| Protein (g) [11] | | Protein: 1.0–1.2 g/kg [11] | | | | | | | | | |
| | ER | 66–79 | 39–47 | 58–70 | 69–82 | 96–115 | 65–78 | 95–114 | 65–78 | | |
| | Actual | 62 | 54 | 22 | 75 | 39 | 11 | 42 | 45 | 44 | 21 |
| | % | 93 | 115 | 37 | 100 | 41 | 17 | 45 | 69 | | |
| Saturated fat (% energy) | | Saturated fat recommended intake: <7% [11] | | | | | | | | | |
| | ER | 13 | 8 | 11 | 13 | 19 | 13 | 18 | 15 | | |
| | EI | 15 | 13 | 13 | 19 | 8 | 4 | 11 | 16 | 12 | 5 |
| | % | 115 | 160 | 118 | 150 | 42 | 31 | 61 | 106 | | |
| Carbohydrate (g) | | Carbohydrates recommended: 50–60% of energy intake [11] | | | | | | | | | |
| | ER | 206–248 | 123–148 | 181–218 | 215–258 | 300–361 | 204–244 | 298–357 | 204–244 | | |
| | Actual | 168 | 116 | 77 | 12 | 136 | 36 | 104 | 97 | 44 | 18 |
| | % | 81 | 94 | 42 | 6 | 45 | 17 | 35 | 48 | | |
| Fibre (g/day) | | Fibre recommended: Men: 30 g/day, Women: 25 g/day [21] | | | | | | | | | |
| | ER | 30 | 25 | 30 | 30 | 30 | 30 | 30 | 30 | | |
| | Actual | 13 | 15 | 6 | 14 | 17 | 6 | 10 | 7 | 11 | 4 |
| | % | 45 | 60 | 20 | 46 | 55 | 20 | 34 | 22 | | |
| Potassium (mmol) a | | Potassium recommended: Limit to 1 mmol/kg/Ideal body weight/day if hyperkalemic [11] | | | | | | | | | |
| | ER | 66 | 35 | 58 | 69 | 96 | 65 | 95 | 65 | | |
| | Actual | 43 | 35 | 13 | 41 | 32 | 17 | 30 | 23 | 29 | 11 |
| | % | 66 | 100 | 23 | 60 | 33 | 26 | 32 | 36 | | |
| Phosphate (mmol) | ER | Phosphorus recommended: 32 mmol/day [19] if hyperphosphatemia [11] | | | | | | | | | |
| | Actual | 29 | 29 | 15 | 33 | 15 | 7 | 18 | 17 | 20 | 9 |
| | % | 91 | 91 | 47 | 103 | 47 | 22 | 56 | 53 | | |
| Sodium (mmol) b | ER | Sodium recommended: <100 mmol day [11] | | | | | | | | | |
| | Actual | 50 | 39 | 18 | 58 | 33 | 16 | 45 | 48 | 38 | 15 |
| | % | 50 | 39 | 18 | 58 | 33 | 16 | 45 | 48 | | |
| Calcium (mg) | ER | Calcium recommended: <1000 mg/day [21] | | | | | | | | | |
| | Actual | 566 | 414 | 309 | 577 | 206 | 194 | 217 | 270 | 344 | 157 |
| | % | 57 | 41 | 31 | 58 | 21 | 19 | 22 | 27 | | |
| Iron (mg) | ER | 8 mg/day [21] | | | | | | | | | |
| | Actual | 9 | 10 | 3 | 12 | 7 | 3 | 9 | 5 | 7 | 3 |
| | % | 113 | 126 | 32 | 145 | 92 | 41 | 116 | 59 | | |
| Zinc (mg) | | Zinc recommended: Men: 14 mg/day, Women: 8 mg/day [21] | | | | | | | | | |
| | ER | 14 | 8 | 14 | 14 | 14 | 14 | 14 | 14 | | |
| | Actual | 6 | 6 | 2 | 12 | 7 | 2 | 8 | 3 | 6 | 3 |
| | % | 46 | 76 | 16 | 89 | 50 | 17 | 55 | 21 | | |
| Folate (µg) | ER | Folate recommended: 400 µg/day [21] | | | | | | | | | |
| | Actual | 295 | 220 | 86 | 313 | 245 | 130 | 310 | 223 | 228 | 83 |
| | % | 74 | 55 | 21 | 78 | 61 | 33 | 77 | 56 | | |
| Vitamin C (mg) | ER | Vitamin C recommended: 45 mg/day [21] | | | | | | | | | |
| | Actual | 32 | 33 | 1 | 20 | 45 | 16 | 22 | 9 | 22 | 14 |
| | % | 71 | 74 | 2 | 44 | 100 | 37 | 49 | 20 | | |

Legend: a—to convert mmol K to mg K—multiply amount by 39. b—to convert mmol Na to g Na- multiple amounts by 0.023.

Phase three identified five themes from the interviews with participants. These themes and the interrelationships between the themes are outlined in Figure 1.

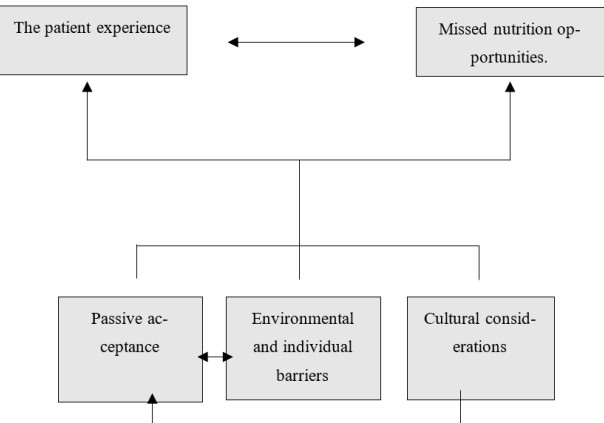

**Figure 1.** The interrelationship between themes is shown below. The themes of passive acceptance, environmental and individual barriers, and cultural considerations influence the overall patient experience and may also lead to missed nutritional opportunities.

The patient experience: Food was considered an important factor that influenced the overall hospital experience of participants. Participants reported that unsatisfactory menu items contributed to reduced energy levels, motivation and a lack of appetite. Nutrition impact symptoms also contributed to recovery and the patient experience.

*"I'm in prison, cause I'm not very hungry"* (Participant 5).

The menu was described as repetitive, lacked flavour and of poor quality which did not meet patient preferences. This in turn decreased patient satisfaction.

*"They look appetising, they look good and you put it in your mouth and it's like ehhh"* (Participant 5).

*"I don't mind fish, but not poached in water . . . It comes out of water and goes back in water"* (Participant 7).

The patients' experiences and dissatisfaction with the menu led to requests for familiar foods from home. Patients commented that this created an unnecessary burden for family members.

*"Sometimes my carer brings the food for me, because I don't like what's on the menu"* (Participant 2).

*" . . . And because he can speak to my mum about what he wants . . . if there's anything wrong with it then he can say this and that. But he doesn't feel like he has that option sometimes with . . . the hospital food"* (Carer of Participant 3).

Missed nutrition opportunities: The undesirable texture of food and unsatisfactory cooking methods resulted in missed opportunities to achieve optimal nutrition for the inpatients interviewed. Other factors such as unfamiliarity with the hospital ordering system, substitution of chosen menu items with non-preferred foods, lack of communication with staff about food preferences and the inability to attend to individual patient needs resulted in missed opportunities to consume food.

*"Because I always ask for something and I don't always get it"* (Participant 2).

Passive acceptance: Participants assumed that the food provided by the hospital was consistent with dialysis-related nutritional needs, despite a high level of dissatisfaction with the options provided.

*"It's probably good for me you know"* (Participant 5).

*"Everything's like healthy you know, what can you do"* (Participant 8).

Some participants expressed a sense of frustration with the food provided. For example, the lack of menu items and the ability to choose menu items that were consistent

with patient preferences were often expressed. Other participants also appeared to passively accept the options provided to them as they were unaware that they could ask for non-standard items.

*"I like juice but they don't like me to have that. They say I'm not supposed to have juice"* (Participant 2).

*"You can't ask for what I want"* (Participant 8).

Environmental and individual barriers: Environmental barriers to optimal intake included lack of staff assistance, and absences from the ward to attend haemodialysis treatment, medical procedures and scans. These absences resulted in frequently missed meals. Individual barriers preventing optimal intake included difficulties opening food packaging, nutrition impact symptoms, and reduced mobility.

*"Yeah so that's the thing, when he comes back after an hour the food might go cold"* (Carer of Participant 1).

*"They're busy okay so no one can really help you to get you up"* (Participant 8).

*"It's not human to be laying in bed and trying to eat"* (Participant 7).

Cultural considerations: Participants reported that the lack of culturally familiar foods on the menu directly impacted patient satisfaction and dietary intake. Some participants perceived that the limited variety reflected the superior nutritional quality of Western meals. Participants also expressed the perception that access to familiar foods would result in improved dietary intake and a faster recovery.

*"He's just not very accepting. He doesn't enjoy the Western sort of cuisine, he just prefers the oriental sort of cuisine that my mum makes"* (Carer of Participant 3).

*"They do different way for our benefit you know"* (Participant 8).

## 4. Discussion

Ensuring that patients receive adequate nutrition in hospital is considered a basic human right [24]. This study found that the inpatient haemodialysis menu provided significantly insufficient energy, fibre, sodium, potassium and vitamin C. Qualitative feedback from patients indicated that there are numerous barriers to intake including the patient experience, passive acceptance, environmental and individual barriers and cultural considerations. These resulted in missed opportunities to optimise nutrient intake, and inadequate oral intake of energy, fibre, sodium, potassium and vitamin C. This also contributed to participants being unable to meet their nutrient needs for phosphate, calcium, zinc and folate, despite the haemodialysis menu analysis determining that the menu provided adequate amounts of these nutrients.

Previous studies have identified that salt-restricted and therapeutic diets offered to inpatients are insufficient in energy, protein and fibre and do not meet patient nutritional requirements [25,26]. This is consistent with our finding of inadequate energy provision from the haemodialysis menu. The inadequate fibre provided by the haemodialysis menu is also similar to results from a previous study for salt-restricted standard menus [25]. This should be of concern to dietitians as it is well known that oral intake is suboptimal among hospital patients and malnutrition can develop rapidly in vulnerable patients with prolonged hospital stays [11,27].

Given the findings from the present study whereby the default haemodialysis menu was significantly lower in sodium and potassium than the evidence-based guidelines, we suggest that the current menu is unnecessarily restrictive and results in suboptimal provision of nutrients to most haemodialysis patients [7–9]. The restrictive nature of the menu resulted in most participants consuming inadequate energy, protein and fibre. There have been limited studies previously published that have quantified the oral intake of inpatients receiving a haemodialysis menu as an inpatient. One study explored the oral intake of elderly inpatients receiving any menu and found that the mean energy intake was

5104 kJ/day [28]. This differed from the findings of our study of haemodialysis inpatients who had an average intake of 3818 kJ/day. Studies conducted on outpatient haemodialysis patients have also found inadequate energy, protein and fibre intake [3,29,30]. Future research exploring the intake of haemodialysis patients over a longer hospital stay would be useful.

Nutrition intake in the present study was influenced by nutrition impact symptoms such as fatigue, dentition and lack of appetite. Patients communicated that simply eating the food seemed like a chore, which has been outlined in previous studies [16,31]. Furthermore, it has been noted that some patients do not receive any visitors to provide them with an external source of nutrition, which further limits nutritional opportunities as well as creating a sense of isolation impacting the overall hospital experience [32]. Interestingly, we found that for those who were able to obtain food from elsewhere, some perceived this to place an unnecessary burden on the family.

An underlying perception that the food provided by the hospital is superior in nutritional quality to cultural or traditional foods was evident amongst participants. However, our findings suggest that this is not the case, nor is it reflected in the dietary intake of the participants. Food is more than the nutrition it provides, and participants voiced that being offered foods that suit their palates will not only help improve their dietary intake but also has the potential to enhance their emotions. Considering these cultural elements may help to enhance the overall patient experience, which has been reported as important in previous studies [12,33,34].

The inability to express autonomy and independence in menu choices was evident in the language used by the participants in this study such as 'they don't like me', which is not consistent with the concept of patient-centred care [33,35]. It was identified that patients were reluctant to communicate their needs with staff as they were 'waiting,' rather than 'asking,' for help. The inability to communicate needs with staff resulted in undermining patient worth. The consequences of this have not yet been explored in the literature. We observed that this potentially results in missed nutrition opportunities that are associated with increased food waste [36] and health care associated costs [37]. Participants were also observed to have a poor understanding of hospital processes pertaining to food ordering. Addressing this deficit may result in improved intake and engagement by patients in their healthcare.

This study has identified that intake of potassium and sodium is below recommended levels amongst the majority of participants. This is in contrast to several studies that found that the intake of potassium, sodium and phosphate exceed recommendations in an outpatient setting [19,30,38]. One possible explanation for this could be that the inpatient menu is overly restrictive and provides limited food choices. In addition, illness-related factors and factors described by patients such as the 'bland' flavour profile of the food provided by the menu could also explain why intake is low. In regard to the phosphate intake being low, the latter reason is most likely the contributing factor as the menu analysis of the haemodialysis diet found that the diet was significantly higher in phosphate than previous restriction recommendations.

Interestingly, this study also identified a significantly lower provision of vitamin C from the menu and an even lower oral intake of vitamin C amongst most participants ($n = 7$). This is an important finding given that patients receiving haemodialysis are at a higher risk of a vitamin C deficiency due to the removal of vitamin C through haemodialysis treatment [39,40]. Thus, if a patient is in hospital for a prolonged time and receiving the haemodialysis menu and regular haemodialysis treatment, they are at a high risk of developing a vitamin C deficiency induced by their hospital admission [39,40]. This finding is not consistent with previous research completed including one study that analysed the vitamin C provision of reduced salt and carbohydrate diets and found that the average provision was adequate compared to 75 mg, which was a higher target than that used in the current study [24]. Given the findings of this study, the reason the haemodialysis diet was found to be significantly inadequate in vitamin C in the present study is potentially due

to the haemodialysis diet being over-restrictive in potassium compared to a standard salt-reduced menu. This phenomenon was also hypothesised in a review article by Handelman et al., 2007 [40]. Thus, a less restrictive approach towards potassium restrictions, as outlined in updated nutritional guidelines for management of chronic kidney disease [11], could resolve this issue. One participant from the current study was able to meet 100% of their vitamin C requirements; however, this participant was ordering additional serves of drained peaches daily compared to the standard default haemodialysis menu. This participant also met 33% of their recommended potassium intake which shows the ability of a menu to optimise vitamin C intake and still meet the potassium restrictions.

Surprisingly, we identified that misinformation and miscommunication of nutrition knowledge regarding the food provided by the hospital exists, which further compromised nutritional intake. However, this information was not provided by a dietitian, which warrants further research into the role of other healthcare professionals delegating incorrect nutrition knowledge to patients. This is imperative as patients perceive healthcare providers as a source that they trust, and they associate advice with improved quality of life [13]. Unexpectedly in this study, an anomaly participant that was much smaller in size compared to other dialysis participants was recruited, and the majority of their nutritional requirements were met from hospital intake only.

The strength of this study is the mixed methods design, which enables us to triangulate intake with the patient experience. To our knowledge, this is the first mixed methods study undertaken with inpatients receiving the haemodialysis inpatient menu. It is recommended that the processes guiding therapeutic diet specifications [18] are reviewed and updated to accommodate the new KDOQI guidelines [11]. This will facilitate improvements in the inpatient hospital menu to ensure adequate provision of all nutrients. It is also recommended that this study is replicated as a multi-centre study to confirm the results obtained and allow for additional exploration of inpatient oral intake and patient perceptions of the haemodialysis menu. This will ensure that hospital-based nutritional interventions are tailored more appropriately to this nutritionally at-risk population.

The limitations include a small sample size recruited from a single study site, which may not be representative of all intake and perceptions of the haemodialysis inpatient population over multiple sites. This study did not collect data on reasons for hospitalisation, medications or comorbid conditions, or analyse their effects on oral intake. A 24 h recall was conducted, which does not highlight changes in intake throughout the duration of admission. The menu analysis may not necessarily be generalisable to other facilities. As with any dietary research, reporting bias can be a limitation. To minimise this, the study applied a strict approach to the analysis of the qualitative data as outlined by Green et al. [26], the cross-checking of all transcripts and coding by a fellow researcher and all final themes being agreed upon by the entire research team. Due to the limitation of COVID-19 lockdowns reducing recruitment, data saturation for all themes was not able to be reached. However, it was clear there was clear repetition of ideas between the eight participants who were reported on in this study.

## 5. Conclusions

Patients receiving haemodialysis treatment are at an increased risk of malnutrition due to their increased nutritional needs and complex nutrition prescription. The present study found that the default current inpatient haemodialysis menu did not enable patients to fully meet their nutrient requirements and that most patients did not eat adequate amounts. The perceptions of patients enabled us to explore the barriers to intake (passive acceptance, environmental and individual barriers and cultural considerations) and the consequences (missed nutrition opportunities). It is recommended that the inpatient haemodialysis menu is optimised in line with evidence-based guidelines for inpatients receiving haemodialysis treatment. Further research exploring patient perceptions of the haemodialysis menu is also recommended in a larger sample size to ensure that data saturation is reached. Further research with a larger sample size is required to more accurately quantify the oral intake

among haemodialysis inpatients. It would also be of interest for future research to explore other factors that may impact oral intake, such as the effect of comorbid conditions and the reason for admission.

**Supplementary Materials:** The following supporting information can be downloaded at: https://www.mdpi.com/article/10.3390/dietetics2030016/s1, Figure S1. Study sample recruitment, Table S1. Interview questions for qualitative phase of the study.

**Author Contributions:** Study design: K.N. and K.L.; data collection: K.N. and F.A.N.; data analysis F.A.N., K.L. and K.N.; writing the manuscript: K.N., F.A.N. and K.L. All authors are in agreement of the manuscript and below statement of author contributions and declare that the content has not been published elsewhere. All authors have read and agreed to the published version of the manuscript.

**Funding:** This research received no external funding.

**Institutional Review Board Statement:** The study was conducted in accordance with the Declaration of Helsinki and approved by the Research Ethics and Governance Information System at Liverpool Hospital (Approval number 2021/ETH00125).

**Informed Consent Statement:** Informed consent was obtained from all subjects involved in the study.

**Data Availability Statement:** Data available at reasonable request.

**Conflicts of Interest:** The authors declare no conflict of interest.

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
