# Peer review of "Nutritional Adequacy and Patient Perceptions of the Hospital Inpatient Haemodialysis Menu: A Mixed Methods Case Series"

_2674-0311, doi:10.3390/dietetics2030016_

Round 1

Reviewer 1 Report

Comments of this reviewer are:

- As this study refers to inpatients, cause of hospitalization should be mentioned as this can affect both appetite and dietary restrictions

- As inpatients are subject to dietary restrictions due to the cause of hospitalization, the hemodialyzed study participants should be compared to other non-hemodialyzed patients with the same causes of hospitalization.

Author Response

Point 1

As this study refers to inpatients, cause of hospitalisation should be mentioned as this can affect appetite and satiety

Response: Thank you for this comment. We are unable to collect this data retrospectively due to ethics limitations that prohibited use of electronic medical records. We have added this potential confounder to the limitations section of this paper: “This study did not collect data on reason for hospitalisation, medications or comborbid conditions or analyse their effect on oral intake.”

Point 2

As inpatients are subject to dietary restrictions due to the cause of hospitalisation, the hemodialyzed study participants should be compared to other no hemodialyzed patients with the same cause of hospitalisation

Response: Thank you for this excellent suggestion. We are are unable to do this for this present study but suggest that this is an original idea for future research.  

Reviewer 2 Report

This paper presents a well-structured study evaluating the nutritional adequacy of the hospital haemodialysis menu, quantifying the dietary intake of hospitalised haemodialysis patients, and exploring patient perceptions of the menu. The methods used, including menu analysis, semi-structured interviews, and dietary intake calculations, are appropriate for the study's aim and objectives.

The results of the study clearly show that the default inpatient haemodialysis menu is inadequate in terms of providing adequate energy, sodium, potassium, vitamin C, and fibre, as compared to reference standards. The finding that half of the participants had inadequate energy and protein intake is particularly concerning. The study also highlights the role of environmental and cultural considerations in impacting the patient experience and limited food intake.

The study's conclusions are well supported by the findings, and the recommendation to optimize the default menu in line with evidence-based guidelines for inpatients is appropriate. However:

1. the study could be strengthened by including a larger sample size of patients and by considering other factors that may impact dietary intake, such as medication use or comorbidities. 

2. it would be helpful to have more information on the evidence-based guidelines used to evaluate the menu's nutritional adequacy.

3. It would also be useful to know what the original dietary prescription was and the difference between what the patients actually consumed.

4. There is an big difference between the BMIs of the patients. It is clear that caloric requirements are different. How is it determined whether they are malnourished or not?

Minor "Fibre recommended Men: 30mg/day, Women: 25mg/day (21)" --> g/day

Fibre recommended Men: 30mg/day, Women: 25mg/day (21)

English is ok

Author Response

 Point 1:

The study could be strengthened by including a larger sample size of patients and considering other factors impacting dietary intake such as medications and comorbidities

Response: The need for a greater sample size and further research into confounding factors is an important aspect to acknowledge and promote within the paper for future research, thus we have made this clearer in our limitations and conclusions section of the paper.

We initially intended to recruit approximately 15 participants in order to reach data saturation for the qualitative aspect of the paper. However, due to COVID hospital lockdowns the recruitment period was reduced considerably and thus only 8 participants were recruited. This is included already within the paper in the methods and discussion section. The intention of this paper was to be the first to describe and quantify the intake of inpatient haemodialysis patients and thus the smaller sample size was appropriate. Given the findings, it would be valuable to explore this further with a greater sample size.

Point 2:

It would be helpful to have more information on the evidence based guidelines used to evaluate the menu nutritional adequacy

Response: We have now clarified this in the paper. For the clarity of readers, we have expanded information on the default diet. The default hospital menu is created in accordance with the state therapeutic diet specifications. The paper now reads:  “The first phase of the study involved analysis of the default items provided on the haemodialysis menu. The therapeutic diet is intended to provide 9500 kilojoules, 90g protein, <100mmol sodium, <70 mmol potassium, <1200mg (38 mmol) phosphate per day.” The defult menu is designed to theretically meet the needs of a 76kg male.”

The standards which the menu was evaluated against were the new KDOQI guidelines for CKD and are explained in the methods section as follows: “Calculations are outlined in Table 1 and were based on the KDOQI guidelines for energy (126kJ/kg), protein (1.2g/kg), sodium (<100mmol/day), potassium (1mmol/kg); the Nutrient Reference Values for fibre (30g/day), iron (8mg/day), zinc (14mg/day), folate (400μg/day), and vitamin C (45mg/day). A phosphate restriction of 32mmol/day was applied”.

 Point 3:

It would be useful to know what the original dietary prescription was and the different between what patients actually consumed

Response: All patients receiving the haemodialysis diet were receiving by default the amounts above ie 9500 kilojoules, 90g protein, <100mmol sodium, <70 mmol potassium, <1200mg (38 mmol) phosphate per day.

 Point 4:

There is a big difference between the BMIs of the patients. It is clear that caloric requirements are different. How was it determined whether they are malnourished or not?

Response: The Patient-Generated Subjective Global Assessment (PGSGA) tool was utilised to assess nutritional status and make the diagnosis of malnutrition in this study. This has been included in the methods section. This tool is commonly used among dietitians to assess malnutrition and is validated for use in renal patients, including haemodialysis patients. This tool uses a variety of parameters to assess malnutrition including: percentage weight loss, adequacy of oral intake and change of oral intake, symptom burden and a physical assessment which has clear aspects that need to be considered. This tool does not actually use BMI to assess malnutrition as it is well documented that obese and overweight people can also be malnourished.

We have made it more clear in the methods section that this tool assessed for malnutrition: “These assessments included the Patient-Generated Subjective Global Assessment (PG-SGA©) to assess nutritional status and determine presence of malnutrition determine the nutritional status of in each participant.”

Point 5:

Fibre recommendation men 30mg day; women 25mg day

Response: This has now been corrected to gram per day.

Reviewer 3 Report

A t test is a statistical test that is used to compare the means of two groups, so it is unclear why it was used to compare average heamodialysis diet with the reference values.

The recommendation is for 76 kg male patient, did you adjust recommendation to body weight of each patients? Half of you patients have body weight lower than 76 kg, so theoretically recommendation for them is lower.

The unit mmol/day is not usually used to present average sodium (and K and P) content of diet. More suitable unit is g/day (according to KDOQI recommendation for sodium is <100 mmol/day OR <2,3 g/day).

Recommendation for saturated fat and carbohydrates should be mention in the Materials and Methods (line 113 to 119).

Author Response

Point 1

A t test is a statistical test that is used to compare the means of two groups, so it unclear why it was used to compare average haemodialysis diet with the reference values.

Thank you for this comment. The one sample t test was used to determine if the average daily provision on the haemodialysis diet was different from the benchmark outlined by the updated KDOQI guidelines.

The manuscript reads: The average daily amount provided on each day of the 14-day haemodialysis menu cycle were sourced and entered into SPSS (version 25; IBM Corporation, Chicago, IL) for energy, protein, fibre, sodium, potassium, phosphate, iron, zinc, folate and vitamin C. Calculations are outlined in Table 1 and were based on the KDOQI guidelines(11) for energy (126kJ/kg), protein (1.2g/kg), sodium (<100mmol/day), potassium (1mmol/kg); the Nutrient Reference Values for fibre (30g/day), iron (8mg/day), zinc (14mg/day), folate (400μg/day), and vitamin C (45mg/day). A phosphate restriction of 32mmol/day was applied(19). A one sample t-test was conducted using SPSS (version 25; IBM Corporation, Chicago, IL) to compare the nutrient profile of the average provision of the haemodialysis default menu to the reference values of a 76kg male according to the updated KDOQI nutrient guidelines(11).”

 Point 2:

The recommendation is for 76 kg male patient, did you adjust recommendation to body weight of each patients? Half of you patients have body weight lower than 76 kg, so theoretically recommendation for them is lower.

Response: We have included further clarification of this point to Table 3 including details of the energy requirements. To respond to the reviewer’s point more explicitly, when comparing daily intake of the menu by the 8 participants to their requirements we multiplied the participant weight by the evidenced-based guideline recommended for energy (105-146kJ/kg).

Point 3:

The unit mmol/day is not usually used to present average sodium (and K and P) content of diet. More suitable unit is g/day (according to KDOQI recommendation for sodium is <100 mmol/day OR <2,3 g/day).

Response: In Australia it is usual dietetic practice to report intake of sodium and potassium as mmol. We have added to the legend of Table 3, the reference for converting mmol to mg of potassium and mmol to g of sodium for clarity for international readers.

Point 4:

Recommendation for saturated fat and carbohydrates should be mention in the Materials and Methods (line 113 to 119).

Response: Thanks you for this comment, the carbohydrate and saturated fat values used have been added to this section, the paper now reads: “Calculations are outlined in Table 1 and were based on the KDOQI guidelines(11) for energy (126kJ/kg), protein (1.2g/kg), saturated fat (<7% of energy intake), carbohydrate (50-60% of energy intake), sodium (<100mmol/day), potassium (1mmol/kg); the Nutrient Reference Values for fibre (30g/day), iron (8mg/day), zinc (14mg/day), folate (400μg/day), and vitamin C (45mg/day).”

Round 2

Reviewer 1 Report

Limitations still exist

Author Response

We have acknowledged this in our limitations section: “The limitations include a small sample size recruited from a single study-site, which may not be representative of all intake and perceptions of the haemodialysis inpatient population over multiple sites. This study did not collect data on reason for hospitalisation, medications or comborbid conditions or analyse their effect on oral intake”.

We also suggest that “It is also recommended that this study be replicated as a multi-centre study to confirm the results obtained and allow for additional exploration of inpatient oral intake and patient perceptions of the haemodialysis menu. This will ensure hospital based nutritional interventions are tailored more appropriately to this nutritionally at-risk population”.

Reviewer 2 Report

The paper adequately discusses the implications of the findings. It highlights that the nutritional inadequacies in the default menu negatively impact the dietary intake and overall experience of haemodialysis inpatients. The identification of passive acceptance of the menu, environmental factors, and cultural considerations as contributors to missed food opportunities provides valuable insights for interventions and improvements.

Good

Author Response

Thank you for the summary. No further changes have been made.

Reviewer 3 Report

The manuscript has been sufficiently improved.

Author Response

Thank you for the feedback